# Signaling Paradigms of H_2_S-Induced Vasodilation: A Comprehensive Review

**DOI:** 10.3390/antiox13101158

**Published:** 2024-09-25

**Authors:** Constantin Munteanu, Cristina Popescu, Andreea-Iulia Vlădulescu-Trandafir, Gelu Onose

**Affiliations:** 1Department of Biomedical Sciences, Faculty of Medical Bioengineering, University of Medicine and Pharmacy “Grigore T. Popa” Iași, 700454 Iași, Romania; 2Neuromuscular Rehabilitation Clinic Division, Clinical Emergency Hospital “Bagdasar-Arseni”, 041915 Bucharest, Romania; andreea-iulia.trandafir@drd.umfcd.ro (A.-I.V.-T.); gelu.onose@umfcd.ro (G.O.); 3Faculty of Medicine, University of Medicine and Pharmacy “Carol Davila”, 020022 Bucharest, Romania

**Keywords:** hydrogen sulfide (H_2_S), vasodilation, protein sulfhydration, ROS scavenging, endothelial function

## Abstract

Hydrogen sulfide (H_2_S), a gas traditionally considered toxic, is now recognized as a vital endogenous signaling molecule with a complex physiology. This comprehensive study encompasses a systematic literature review that explores the intricate mechanisms underlying H_2_S-induced vasodilation. The vasodilatory effects of H_2_S are primarily mediated by activating ATP-sensitive potassium (K_ATP) channels, leading to membrane hyperpolarization and subsequent relaxation of vascular smooth muscle cells (VSMCs). Additionally, H_2_S inhibits L-type calcium channels, reducing calcium influx and diminishing VSMC contraction. Beyond ion channel modulation, H_2_S profoundly impacts cyclic nucleotide signaling pathways. It stimulates soluble guanylyl cyclase (sGC), increasing the production of cyclic guanosine monophosphate (cGMP). Elevated cGMP levels activate protein kinase G (PKG), which phosphorylates downstream targets like vasodilator-stimulated phosphoprotein (VASP) and promotes smooth muscle relaxation. The synergy between H_2_S and nitric oxide (NO) signaling further amplifies vasodilation. H_2_S enhances NO bioavailability by inhibiting its degradation and stimulating endothelial nitric oxide synthase (eNOS) activity, increasing cGMP levels and potent vasodilatory responses. Protein sulfhydration, a post-translational modification, plays a crucial role in cell signaling. H_2_S S-sulfurates oxidized cysteine residues, while polysulfides (H_2_Sn) are responsible for S-sulfurating reduced cysteine residues. Sulfhydration of key proteins like K_ATP channels and sGC enhances their activity, contributing to the overall vasodilatory effect. Furthermore, H_2_S interaction with endothelium-derived hyperpolarizing factor (EDHF) pathways adds another layer to its vasodilatory mechanism. By enhancing EDHF activity, H_2_S facilitates the hyperpolarization and relaxation of VSMCs through gap junctions between endothelial cells and VSMCs. Recent findings suggest that H_2_S can also modulate transient receptor potential (TRP) channels, particularly TRPV4 channels, in endothelial cells. Activating these channels by H_2_S promotes calcium entry, stimulating the production of vasodilatory agents like NO and prostacyclin, thereby regulating vascular tone. The comprehensive understanding of H_2_S-induced vasodilation mechanisms highlights its therapeutic potential. The multifaceted approach of H_2_S in modulating vascular tone presents a promising strategy for developing novel treatments for hypertension, ischemic conditions, and other vascular disorders. The interaction of H_2_S with ion channels, cyclic nucleotide signaling, NO pathways, ROS (Reactive Oxygen Species) scavenging, protein sulfhydration, and EDHF underscores its complexity and therapeutic relevance. In conclusion, the intricate signaling paradigms of H_2_S-induced vasodilation offer valuable insights into its physiological role and therapeutic potential, promising innovative approaches for managing various vascular diseases through the modulation of vascular tone.

## 1. Introduction

Traditional therapies using sapropelic mud (peloid), a complex mixture of fine mineral particles, organic matter, and water [1], typically sourced from specific geographical locations [2] or sulfurous mineral waters, have been used for millennia for their health benefits [3]. Modern scientific advancements have begun to elucidate the molecular and biochemical mechanisms underlying their effects, emphasizing the role of H_2_S [4]. The clinical applications of therapeutic sapropelic mud and sulfurous mineral waters are diverse, spanning the alleviation of symptoms associated with rheumatic diseases, skin conditions, and respiratory disorders [5]. The mineral content of peloids also includes elements such as calcium, magnesium, and potassium, while their organic matter can contain humic substances, amino acids, and other bioactive compounds [6,7]. The physical properties of these muds, such as texture and thermal capacity, also play a significant role in their therapeutic applications. Mud therapy, or pelotherapy, involves applying mud packs or baths to treat musculoskeletal pain, inflammation, and skin problems [8].

Sulfurous mineral waters, rich in dissolved sulfur compounds, including hydrogen sulfide (H_2_S) and sulfate (SO_4_^2−^), are another potent therapeutic resource [9]. These waters are often found in geothermal regions influenced by volcanic activity or subterranean mineral deposits [10]. The concentration of H_2_S, along with other minerals such as calcium, magnesium, and sodium, contributes to the therapeutic properties of these waters [11]. Additionally, these waters contain essential ions and trace elements like iron, manganese, and zinc, which further enhance their health benefits [12].

The physicochemical properties of therapeutic sapropelic mud and sulfurous mineral waters are directly linked to their health benefits. For instance, therapeutic mud’s high specific heat capacity and low thermal conductivity make it effective for heat therapy, aiding in pain relief, muscle relaxation, and improved circulation [13,14]. The pH, redox potential, viscosity, and mineral content of mud and sulfurous waters influence their ability to support skin health, reduce oxidative stress, and promote wound healing or other health issues [15].

H_2_S, pivotal in the therapeutic mechanisms of mud and sulfurous mineral waters [16], has recently gained attention as a significant gaseous signaling molecule in biomedical research, particularly due to its role in mediating vasodilation, regulating blood pressure and vascular tone, and enhancing endothelial function [17,18]. Known for its vasodilatory properties, H_2_S improves blood flow and reduces blood pressure, making it a valuable therapeutic agent [19]. Advances in molecular biology and biochemistry have provided more insights into the mechanisms by which H_2_S exerts its effects [20]. Recent studies have identified several signaling paradigms through which H_2_S exerts its vasodilatory effects [21]. One significant pathway involves the activation of ATP-sensitive potassium (K_ATP) channels in vascular smooth muscle cells, leading to membrane hyperpolarization and relaxation of the blood vessels [22]. Additionally, H_2_S has been shown to increase the production of nitric oxide (NO) by activating endothelial NO synthase (eNOS), which further contributes to vasodilation [23]. Another critical mechanism includes the modulation of calcium homeostasis in smooth muscle cells by inhibiting voltage-gated calcium channels and reducing intracellular calcium concentrations, which promotes vascular relaxation [19,24].

Endogenously produced in the human body through enzymatic pathways involving cystathionine β-synthase (CBS), cystathionine γ-lyase (CSE), and 3-mercaptopyruvate sulfurtransferase (3-MST) [25]. H_2_S is present in various tissues, including the brain, liver, kidneys, and blood vessels [26]. CBS, found primarily in the brain, liver, and kidneys, regulated by S-adenosylmethionine (SAM) and redox state, catalyzes the conversion of homocysteine and serine into cystathionine, with H_2_S production occurring as a side reaction. CSE, predominantly located in the liver, kidneys, and vascular tissues, regulated by pyridoxal 5′-phosphate (PLP) and redox level, converts cystathionine into cysteine, ammonia, and α-ketobutyrate, producing H_2_S from cysteine and homocysteine. 3-MST, present in the mitochondria and cytoplasm of various tissues, produces H_2_S by transferring a sulfur atom from 3-mercaptopyruvate, with its activity influenced by substrate availability and oxidative stress. Once produced, H_2_S interacts with multiple molecular targets, such as ion channels, including ATP-sensitive potassium channels [27] and voltage-gated calcium channels [24], enzymes [28], and transcription factors such as nuclear factor kappa B (NF-κB) [29] and hypoxia-inducible factor-1α (HIF-1α) [30,31].

H_2_S also plays a crucial role in modulating oxidative stress [32] and inflammation [33]. It enhances the activity of antioxidant enzymes like superoxide dismutase (SOD) [34] and glutathione peroxidase (GPx), neutralizing reactive oxygen species (ROS) and protecting cells from oxidative damage [35]. The therapeutic applications of H_2_S are actively being explored in various clinical contexts. In cardiovascular diseases, H_2_S donors and H_2_S-releasing compounds are being investigated for their potential to treat hypertension, heart failure, and ischemic heart disease by improving vascular function and reducing oxidative stress [36].

The primary objective of this review is to provide a comprehensive and detailed analysis of the molecular mechanisms through which H_2_S mediates the therapeutic effects of mud and sulfurous mineral waters, inducing vasodilation. By integrating current biochemical and molecular research, this review aims to elucidate how H_2_S interacts with various cellular and molecular targets to exert its health benefits.

## 2. Methodology for Systematic Literature Review

The systematic literature review (SLR) aims to comprehensively investigate the signaling paradigms of hydrogen sulfide (H_2_S)-induced vasodilation. The central research question guiding this review is: “What are the molecular and cellular signaling mechanisms through which H_2_S induces vasodilation in various vascular systems?” This question is structured to delve into the intricate pathways involved in H_2_S signaling, examining its interactions with different molecular targets, the cellular responses triggered, and the resultant physiological effects on vasodilation.

Specific inclusion and exclusion criteria were established to ensure the review is focused and relevant. The inclusion criteria encompass studies involving mammalian models, including humans, where H_2_S-induced vasodilation was evaluated. These studies must investigate the molecular and cellular mechanisms underlying H_2_S signaling related to vasodilation. Outcomes of interest include those related to vasodilation, such as changes in vascular tone, blood pressure, and endothelial function. The review includes experimental studies across in vivo, in vitro, and ex vivo designs, as well as clinical trials, without restriction on publication date to capture the full scope of relevant research. Only articles published in English were considered. Conversely, studies focusing on H_2_S effects unrelated to vasodilation, research involving non-mammalian models, or plant studies were excluded. Additionally, review articles, books, editorials, letters, and conference abstracts without full experimental data were not included in this review.

The following electronic databases were searched: PubMed, Web of Science, and Scopus. The search strategy, limited to articles published from 2000 to the present, included the following syntax combination: “hydrogen sulfide” OR “H_2_S” AND “vasodilation” OR “vascular regulation” (Table 1). Additionally, reference lists of selected articles and relevant reviews were hand-searched to identify further studies. Finally, manually searching for references on Google Scholar was also conducted to ensure comprehensive coverage and fill explanatory gaps.

Data extraction was performed using a standardized form to capture essential information from each included study. Key data extracted included study identification details (such as author, year of publication, journal, and study location), study design (e.g., in vivo, in vitro, ex vivo, and clinical), population characteristics (species, cell type, or vascular tissue used), intervention specifics (concentration or method of H_2_S administration and other relevant experimental conditions), signaling mechanisms (detailed description of molecular and cellular pathways, including key molecules like nitric oxide (NO), cGMP, KATP channels, and other signaling intermediaries), outcomes related to vasodilation (such as changes in blood vessel diameter, blood pressure, or specific signaling markers), and results (summary of main findings, including quantitative data like effect sizes, *p*-values, and confidence intervals). Methodological quality indicators, such as randomization, blinding, and the use of controls, were also extracted.

Given the anticipated heterogeneity in study designs, populations, and outcomes, the data synthesis primarily involved a narrative synthesis. The synthesis also involved thematic analysis, identifying common themes and patterns in the mechanisms through which H_2_S induces vasodilation and highlighting any contradictions or gaps in the evidence. It should be emphasized that related literature is still scarce, thus warranting future knowledge quests and gains in acquiring knowledge in this field.

## 3. Hydrogen Sulfide Signaling Pathways Paradigms in Vasodilation

H_2_S has emerged as a critical endogenous gasotransmitter with profound effects on vascular function, particularly in the modulation of vasodilation [19]. This section delves into the specific signaling paradigms through which H_2_S induces vasodilation, highlighting its multifaceted mechanisms within the vascular system. H_2_S promotes vascular smooth muscle relaxation through several distinct pathways, including activating ATP-sensitive potassium channels (K_ATP), inhibiting calcium influx, and modulation of cyclic nucleotide signaling. Additionally, H_2_S interacts synergistically with nitric oxide (NO) signaling, enhancing NO bioavailability and promoting vasodilation through direct and indirect mechanisms. Furthermore, H_2_S contributes to endothelial-dependent hyperpolarization and influences the production of other vasodilatory agents like prostacyclin, showcasing its role in fine-tuning vascular tone [36].

### 3.1. Persulfidation Signaling

Persulfidation, also known as S-sulfhydration, is a critical post-translational modification that modulates the activity of numerous proteins by altering cysteine residues. This process is central to many of the biological effects attributed to hydrogen sulfide (H_2_S). It is important to clarify, as pointed out by Kabil et al. (2014) [37], that H_2_S itself does not interact with reduced cysteine residues directly. Instead, H_2_S targets oxidized cysteine residues, such as those in the form of sulfenic acids (-SOH), sulfinic acids (-SO_2_H), or disulfides (-S-S-), converting them into persulfides (-SSH). This reaction leads to the formation of persulfide groups, which can significantly alter the target proteins’ activity, stability, and localization. Through this modification, H_2_S plays a crucial role in redox signaling and regulating oxidative stress responses. However, it is crucial to distinguish that only polysulfides, such as H_2_Sn (where n > 1) or preformed persulfides, are capable of modifying reduced cysteine residues (free thiols, -SH). Polysulfides containing multiple sulfur atoms are more reactive than H_2_S and can donate sulfur atoms to reduced cysteines, forming persulfides directly on these residues. This distinction highlights the specific roles of different sulfur species in cellular processes. H_2_S serves as an upstream modulator, facilitating the conversion of oxidized cysteines into persulfides. In contrast, polysulfides and existing persulfides mediate further modification of reduced thiol groups, contributing to broader cellular functions, including vasodilation, cytoprotection, and antioxidant defense.

During persulfidation, H_2_S_n_ and other per- and poly-sulfides donate a sulfur atom to the cysteine residue’s thiol (-SH) group within a protein, forming a persulfide (-SSH) group [37]. This reaction typically occurs under physiological conditions and can be catalyzed or facilitated by various cellular factors. This chemical modification can significantly influence the target protein’s function by altering its activity, stability, and subcellular localization [26,32].

Persulfidation can enhance or inhibit the enzymatic activity of proteins. For example, persulfidation of glyceraldehyde-3-phosphate dehydrogenase (GAPDH) increases its glycolytic activity, while persulfidation of protein tyrosine phosphatases (PTPs) can inhibit their activity, thereby affecting downstream signaling pathways. Introducing a persulfide group can also affect the structural stability of a protein. This modification may either stabilize the protein, protect it from degradation, or induce conformational changes that lead to its destabilization and subsequent degradation. Persulfidation can influence the localization of proteins within the cell. For instance, it may promote the translocation of proteins to specific cellular compartments, such as the nucleus, mitochondria, or membrane, modulating their functional roles in different cellular contexts [38].

Persulfidation is a critical mechanism by which H_2_S modulates various signaling pathways. By modifying key signaling proteins, H_2_S can influence apoptosis, inflammation, and cellular stress responses. Persulfidation also plays a role in maintaining redox homeostasis. It can protect cysteine residues from irreversible oxidation under oxidative stress conditions, thereby preserving protein function [39]. In the context of vascular function, persulfidation of proteins involved in vasodilation, such as K_ATP channels and endothelial nitric oxide synthase (eNOS), enhances their activity, leading to increased production of vasodilatory signals and improved blood flow [19].

### 3.2. K_ATP Channel Activation Paradigm

The K_ATP channel activation paradigm is one of the primary mechanisms by which H_2_S induces vasodilation. H_2_S directly activates ATP-sensitive potassium (K_ATP) channels located on the membrane of vascular smooth muscle cells (VSMCs). These channels are sensitive to the intracellular levels of ATP and are typically closed under normal conditions when ATP levels are sufficient to maintain membrane potential. When H_2_S interacts with these K_ATP channels, it causes them to open, leading to the efflux of potassium ions (K⁺) from the smooth muscle cells. This outward movement of K⁺ results in the hyperpolarization of the cell membrane [40].

Membrane hyperpolarization decreases the activity of voltage-dependent calcium channels, reducing the influx of calcium ions (Ca^2+^) into the cells. Lower intracellular calcium levels lead to the relaxation of vascular smooth muscle cells, causing the blood vessels to dilate. This vasodilation reduces vascular resistance and increases blood flow, contributing to blood pressure regulation [41]. H_2_S can directly influence vascular tone through ion channel modulation. This mechanism is particularly significant in conditions where vascular tone must be carefully controlled, such as regulating systemic blood pressure or responding to ischemic conditions [23].

### 3.3. cGMP Pathway Activation Paradigm

H_2_S inhibits phosphodiesterase (PDE) enzymes, increasing smooth muscle cells’ cyclic guanosine monophosphate (cGMP) levels. Elevated cGMP activates protein kinase G (PKG), which reduces intracellular calcium levels and causes smooth muscle relaxation and vasodilation. The cGMP pathway activation paradigm is another key mechanism through which H_2_S induces vasodilation. H_2_S significantly modulates cyclic (cGMP) levels within VSMCs. This process begins with H_2_S inhibiting the activity of PDE enzymes, particularly PDE5, which is responsible for cGMP breakdown into its inactive form. By inhibiting PDE, H_2_S leads to an accumulation of cGMP within the VSMCs. PKG phosphorylates several downstream targets, including ion channels and contractile proteins, leading to a reduction in intracellular calcium levels and a decrease in the sensitivity of the contractile apparatus to calcium [42].

The activation of PKG by elevated cGMP levels results in the relaxation of vascular smooth muscle cells. This relaxation is due to both a reduction in calcium influx and a decrease in calcium sensitivity, effectively leading to vasodilation. The dilation of blood vessels reduces vascular resistance and facilitates blood flow, contributing to the regulation of blood pressure and enhancing tissue perfusion. The activation of the cGMP pathway by H_2_S underscores its ability to modulate vascular tone through intracellular signaling cascades that influence calcium handling and muscle contraction. This mechanism is particularly relevant in cardiovascular therapies for treating conditions such as hypertension, where enhancing cGMP signaling can provide significant therapeutic benefits [43].

### 3.4. Endothelial Nitric Oxide Synthase (eNOS) Activation Paradigm

The endothelial nitric oxide synthase (eNOS) activation paradigm highlights the synergistic relationship between H_2_S and nitric oxide (NO) in promoting vasodilation [44]. H_2_S enhances NO production by directly stimulating eNOS activity in endothelial cells. eNOS is the enzyme responsible for synthesizing NO from L-arginine, a crucial process for maintaining vascular homeostasis. H_2_S facilitates eNOS activation through several mechanisms, including post-translational modifications such as S-sulfhydration, which can enhance eNOS enzymatic activity. Additionally, H_2_S can increase the availability of eNOS cofactors, such as tetrahydrobiopterin (BH_4_), thereby boosting NO production. The NO produced by eNOS diffuses from endothelial cells into the adjacent VSMCs. Once inside the VSMCs, NO activates the soluble guanylate cyclase (sGC), which catalyzes the conversion of guanosine triphosphate (GTP) to cGMP-. The elevation of cGMP levels leads to the activation of protein kinase G (PKG), which, in turn, reduces intracellular calcium levels and decreases the contractility of the smooth muscle cells [45].

The increased production of NO due to H_2_S-induced eNOS activation results in potent vasodilation. This process lowers vascular resistance and improves blood flow, contributing to blood pressure regulation and overall vascular health. The interplay between H_2_S and NO in this paradigm emphasizes the complex and cooperative nature of gasotransmitter signaling in the cardiovascular system. The eNOS activation by H_2_S represents a critical pathway through which H_2_S exerts its vasodilatory effects, particularly in conditions where enhanced NO signaling is beneficial, such as managing hypertension, atherosclerosis, and other cardiovascular diseases. This paradigm underscores the potential therapeutic strategies that could be developed by targeting the H_2_S–NO signaling axis to modulate vascular tone effectively [46].

### 3.5. Calcium Signaling Modulation Paradigm

The calcium signaling modulation paradigm is a crucial mechanism through which H_2_S induces vasodilation. Calcium ions (Ca^2+^) play a central role in the contraction of VSMCs. The contraction process is primarily driven by the influx of Ca^2+^ through L-type voltage-dependent calcium channels (VDCCs) on the plasma membrane of VSMCs. This influx triggers a cascade of events that lead to muscle contraction and, consequently, vasoconstriction. H_2_S modulates this process by inhibiting L-type calcium channels, thereby reducing the influx of Ca^2+^ into the smooth muscle cells [41,47].

Additionally, H_2_S may enhance the sequestration of Ca^2+^ into the sarcoplasmic reticulum or promote the extrusion of Ca^2+^ from the cell, further decreasing the intracellular Ca^2+^ concentration. By lowering the availability of Ca^2+^ within the cytoplasm, H_2_S disrupts the calcium–calmodulin complex formation, essential for activating myosin light chain kinase (MLCK), the enzyme that initiates smooth muscle contraction. The inhibition of calcium entry and the reduction in intracellular Ca^2+^ levels lead to the relaxation of VSMCs, resulting in vasodilation [48]. This decrease in vascular tone lowers blood pressure and enhances blood flow, contributing to improved tissue perfusion. The calcium signaling modulation by H_2_S highlights its role as a regulator of vascular tone through the direct manipulation of ion channels and intracellular calcium dynamics. This mechanism is particularly relevant in conditions such as hypertension and ischemia, where excessive vasoconstriction needs to be mitigated. By targeting calcium signaling pathways, H_2_S offers a potential therapeutic avenue for managing these cardiovascular conditions and maintaining vascular health [23,49].

### 3.6. Redox Signaling and Antioxidant Effects Paradigm

The redox signaling and antioxidant effects paradigm emphasizes H_2_S’s role in modulating oxidative stress and its subsequent impact on vasodilation. Oxidative stress, characterized by the overproduction of reactive oxygen species (ROS), significantly contributes to endothelial dysfunction and impaired vasodilation [50]. ROS can rapidly react with nitric oxide (NO), reducing its bioavailability and forming peroxynitrite (ONOO^−^), a reactive nitrogen species that further exacerbates oxidative damage and vasoconstriction [51]. H_2_S functions as an endogenous antioxidant, directly scavenging ROS such as superoxide (O_2_^−^) and hydrogen peroxide (H_2_O_2_), thereby reducing oxidative stress within the vascular system [52]. Additionally, H_2_S upregulates the expression and activity of various antioxidant enzymes, including superoxide dismutase (SOD), catalase, and glutathione peroxidase. These enzymes play critical roles in neutralizing ROS and maintaining redox balance. By mitigating oxidative stress, H_2_S preserves the bioavailability of NO, allowing it to exert its vasodilatory effects through activating sGC and the subsequent increase in cGMP levels [53].

The antioxidant effects of H_2_S led to enhanced NO-mediated vasodilation, reduced vascular inflammation, and improved endothelial function. Reducing oxidative stress and preserving NO bioavailability contributes to lower vascular resistance, improved blood flow, and better blood pressure regulation. The redox signaling and antioxidant effects paradigm underscores the importance of H_2_S in protecting the vascular system from oxidative damage. It highlights its potential therapeutic role in conditions associated with oxidative stress, such as hypertension, atherosclerosis, and other cardiovascular diseases. Through its dual role as an antioxidant and a modulator of redox signaling, H_2_S plays a crucial part in maintaining vascular homeostasis and promoting vasodilation [37].

### 3.7. Interaction with Other Gasotransmitters Paradigm

H_2_S interacts with other gasotransmitters, such as nitric oxide (NO) and carbon monoxide (CO), either synergistically or antagonistically, to regulate vascular tone. This crosstalk fine-tunes the vasodilatory response [54]. One of the most notable interactions occurs between H_2_S and NO. The synergistic effect of H_2_S with NO in vasculature was first reported by Hosoki et al. (1997), who demonstrated that the two gases work together to relax vascular smooth muscle [55]. Later, Whiteman et al. (2006) suggested that the chemical interaction of both molecules produces nitrosothiol [56]. More recent studies have proposed other products, including nitroxyl (HNO) by Eberhardt et al. (2014) [57], SSNO- by Cortese-Krott et al. (2014) [58], and polysulfides H_2_S_n_ by Miyamoto et al. (2017) [59].

H_2_S enhances the vasodilatory effects of NO by several mechanisms: H_2_S inhibits the oxidative degradation of NO by reducing ROS levels, thereby preserving NO bioavailability; H_2_S can upregulate eNOS activity, leading to increased NO production in endothelial cells; both H_2_S and NO contribute to the activation of sGC in VSMCs, resulting in a synergistic increase in cyclic guanosine monophosphate (cGMP) levels and enhanced vasodilation [23,60].

The interaction between H_2_S and CO, another gasotransmitter, also plays a role in vascular regulation [61]. Although CO and H_2_S can have opposing effects, they can work together to balance vascular tone. For example, while CO may cause vasoconstriction in certain contexts, its interaction with H_2_S can modulate this effect, leading to a more controlled vascular response [62].

The interplay between H_2_S and other gasotransmitters like NO and CO results in a finely tuned regulation of vasodilation, ensuring that vascular tone is appropriately adjusted to meet the physiological demands of the body. This crosstalk enhances the overall effectiveness of vasodilatory signaling, contributing to better blood flow regulation, reduced blood pressure, and protection against vascular dysfunction [63].

### 3.8. Endothelial-Dependent Hyperpolarization (EDH) Paradigm

The Endothelial-Dependent Hyperpolarization (EDH) paradigm highlights a crucial mechanism through which H_2_S contributes to vasodilation, particularly in small resistance arteries. EDH refers to the process by which blood vessel endothelial cells (ECs) cause hyperpolarization in the underlying VSMCs, leading to vasodilation. This mechanism is especially important in smaller arteries, where regulating vascular tone is critical for maintaining blood flow and pressure [64].

H_2_S plays a significant role in enhancing EDH-mediated responses through several mechanisms. First, H_2_S enhances the activity of potassium channels in endothelial cells, particularly small and intermediate conductance calcium-activated potassium channels. The activation of these channels leads to the efflux of potassium ions (K^+^) from endothelial cells, causing the endothelial cell membrane to hyperpolarize. This hyperpolarization is a key step in the EDH process and directly contributes to the relaxation of VSMCs [65,66].

Moreover, the hyperpolarization of endothelial cells is transmitted to the adjacent VSMCs through myoendothelial gap junctions—specialized connections that allow direct electrical and chemical communication between these cells. The hyperpolarization of VSMCs reduces the activity of voltage-dependent calcium channels, decreasing intracellular calcium levels and promoting muscle relaxation. This electrical coupling via gap junctions is essential for the coordinated regulation of vascular tone across different segments of the vascular system [67].

In addition to these direct effects, H_2_S may also influence the production or action of other endothelium-derived hyperpolarizing factors (EDHFs), such as epoxyeicosatrienoic acids (EETs) or hydrogen peroxide (H_2_O_2_). Enhanced production of endothelium-derived hyperpolarizing factors and reduced production of hydrogen peroxide (H_2_O_2_) induced by H_2_S contribute to the hyperpolarization of VSMCs and enhance vasodilation. The interaction between H_2_S and these EDHFs highlights the multifaceted role of H_2_S in regulating vascular function [68].

### 3.9. Interaction with Prostacyclin Signaling Paradigm

H_2_S can enhance prostacyclin (PGI_2_) production, which is a potent vasodilator that endothelial cells produce. Prostacyclin, a member of the prostanoid family of lipid mediators, plays a crucial role in maintaining vascular homeostasis by promoting vasodilation, inhibiting platelet aggregation, and protecting against thrombosis [49]. H_2_S can upregulate the prostacyclin synthesis in endothelial cells by stimulating cyclooxygenase (COX) enzymes, particularly COX-2, which catalyze the conversion of arachidonic acid to prostacyclin. H_2_S may enhance COX-2 expression or activity, leading to increased prostacyclin production. This upregulation of prostacyclin synthesis is a key mechanism through which H_2_S contributes to vascular relaxation [69,70].

Prostacyclin exerts its vasodilatory effects by binding to IP receptors (prostacyclin receptors) on the surface of VSMCs. Upon binding, these G protein-coupled receptors activate adenylate cyclase, increasing cyclic adenosine monophosphate (cAMP) levels within the cells. Elevated cAMP levels activate protein kinase A (PKA), which phosphorylates and inactivates myosin light chain kinase (MLCK). This inactivation reduces the sensitivity of the contractile apparatus to calcium, leading to smooth muscle relaxation and subsequent vasodilation [71].

In addition to stimulating prostacyclin production, H_2_S works synergistically with prostacyclin to enhance vasodilation. By promoting adenylate cyclase activity and increasing cAMP levels, H_2_S amplifies the signaling cascade initiated by prostacyclin, resulting in a more pronounced vasodilatory response. This synergistic effect between H_2_S and prostacyclin contributes to a powerful reduction in vascular resistance and improved blood flow. The interaction between H_2_S and prostacyclin is particularly important in protecting against conditions involving vascular dysfunction, such as hypertension, atherosclerosis, and thrombosis [72].

### 3.10. Hypoxia Response Paradigm

The Hypoxia Response Paradigm outlines the role of H_2_S in modulating vascular responses to low oxygen conditions (hypoxia). Hypoxia triggers a complex set of cellular responses to restore oxygen supply and maintain cellular function [73]. H_2_S is increasingly recognized as a key modulator in the adaptive response to hypoxia [63,74,75].

One of the primary mechanisms through which H_2_S influences the hypoxia response is stabilizing hypoxia-inducible factors (HIFs), particularly HIF-1α [76]. Under normoxic conditions, HIF-1α is rapidly degraded by prolyl hydroxylases (PHDs), which require oxygen to function. During hypoxia, PHD activity decreases, allowing HIF-1α to accumulate and translocate to the nucleus, activating as a transcription factor of angiogenesis, erythropoiesis, and glycolysis genes. H_2_S can inhibit PHD activity, enhancing HIF-1α stability even under mild hypoxia [77]. H_2_S directly contributes to vasodilation in hypoxic conditions by activating ATP-sensitive potassium (K_ATP) channels and inhibiting L-type calcium channels in VSMCs, leading to decreased intracellular calcium levels and muscle relaxation. This vasodilation is crucial in hypoxia as it helps increase blood flow to tissues, enhancing oxygen delivery [78].

H_2_S also interacts with mitochondrial respiratory complexes, particularly complex IV (cytochrome c oxidase), modulating oxygen consumption and promoting the adaptation of cellular metabolism to low oxygen availability. This effect helps optimize the efficiency of cellular respiration under hypoxic conditions, reducing oxidative stress and preserving ATP production [79].

### 3.11. H_2_S Interaction with Transcription Factors

H_2_S influences several nuclear factor signaling pathways that regulate gene expression, particularly in vasodilation. These pathways involve various transcription factors that respond to changes in cellular redox states, inflammation, and other stress signals [53,80].

H_2_S enhances the activity of NRF2 [81], a key transcription factor that regulates the expression of antioxidant and cytoprotective genes. By promoting the S-sulfhydration of Keap1, which normally inhibits NRF2, H_2_S facilitates the release and nuclear translocation of NRF2. In the nucleus, NRF2 binds to antioxidant response elements (ARE) in the promoters of target genes, leading to the upregulation of antioxidant enzymes such as heme oxygenase-1 (HO-1), superoxide dismutase (SOD), and glutathione peroxidase [82]. Activating NRF2 by H_2_S reduces oxidative stress in endothelial cells, preserving nitric oxide (NO) bioavailability. NO is a critical vasodilator that relaxes vascular smooth muscle cells (VSMCs) by increasing cyclic GMP levels. By reducing oxidative inactivation of NO, H_2_S indirectly supports sustained vasodilation, contributing to vascular health and blood pressure regulation [54].

H_2_S stabilizes HIF-1α, a transcription factor that responds to low oxygen levels (hypoxia) by inducing the expression of genes that facilitate adaptation to hypoxia, including those involved in angiogenesis, erythropoiesis, and glycolysis. H_2_S inhibits prolyl hydroxylases (PHDs), which mark HIF-1α for degradation under normoxic conditions. This inhibition allows HIF-1α to accumulate and translocate to the nucleus, activating hypoxia-responsive genes. HIF-1α activation by H_2_S promotes the expression of vascular endothelial growth factor (VEGF), stimulating the formation of new blood vessels and enhancing blood flow [83,84].

H_2_S modulates the activity of NF-κB, a transcription factor that regulates the expression of pro-inflammatory cytokines, adhesion molecules, and other mediators of inflammation. H_2_S can inhibit NF-κB activation by S-sulfhydrating key cysteine residues on the IκB kinase (IKK) complex, preventing the phosphorylation and degradation of IκB, the inhibitor of NF-κB. As a result, NF-κB remains sequestered in the cytoplasm and inactive. By inhibiting NF-κB, H_2_S reduces the expression of inflammatory cytokines and adhesion molecules in the endothelium, mitigating vascular inflammation. This anti-inflammatory effect helps maintain endothelial function and NO availability, promoting vasodilation. Additionally, reducing inflammation prevents the development of atherosclerotic plaques, which can impair blood flow and contribute to hypertension [85,86].

H_2_S activates sirtuins, particularly SIRT1 and SIRT3, which are NAD⁺-dependent deacetylases involved in regulating cellular stress responses, metabolism, and longevity. SIRT1 deacetylates and activates eNOS, enhancing NO production. SIRT3, on the other hand, promotes mitochondrial function and reduces oxidative stress by deacetylating and activating superoxide dismutase 2 (SOD2) and other mitochondrial proteins. By activating SIRT1, H_2_S enhances NO production, directly contributing to vasodilation. SIRT3-mediated reduction in mitochondrial ROS further preserves NO bioavailability, supporting sustained vasodilation. These effects underscore the role of H_2_S in maintaining vascular health and protecting against conditions like hypertension and atherosclerosis [87].

By modulating the activity of NRF2, HIF-1α, NF-κB, and sirtuins, H_2_S enhances NO production, reduces oxidative stress and inflammation, and supports endothelial function. These mechanisms collectively promote vasodilation, highlighting the therapeutic potential of H_2_S [32].

## 4. Therapeutic Mechanisms of H_2_S Resulted from Inducing Vasodilation

H_2_S has garnered significant attention for its therapeutic potential, particularly its ability to induce vasodilation (Figure 1). The vasodilatory effects of H_2_S are crucial for maintaining vascular homeostasis and have important therapeutic implications for various diseases [88].

One of the primary therapeutic mechanisms of H_2_S is blood pressure regulation. H_2_S-induced vasodilation leads to the relaxation of VSMCs, widening blood vessels. This reduction in vascular resistance lowers systemic blood pressure, making H_2_S a potential therapeutic agent for managing hypertension. By reducing blood pressure, H_2_S can help prevent the onset of hypertension-related complications such as stroke, myocardial infarction, and chronic kidney disease. It offers an alternative or complementary approach to traditional antihypertensive therapies, particularly in patients who are resistant to conventional treatments [89].

Another critical role of H_2_S is in protecting against ischemia-reperfusion injury. During ischemia, tissues experience a lack of oxygen and nutrients, leading to cellular damage. Reperfusion, or the restoration of blood flow, can further exacerbate this damage through oxidative stress. H_2_S-induced vasodilation improves blood flow during reperfusion, reducing the extent of injury by restoring oxygen and nutrient delivery more gradually and evenly. In conditions such as myocardial infarction, stroke, or organ transplantation, where an ischemia-reperfusion injury is a major concern, H_2_S can protect tissues from damage. Its ability to modulate blood flow and reduce oxidative stress during reperfusion makes it a promising therapeutic candidate for minimizing tissue injury and improving recovery outcomes [90,91,92].

H_2_S also exhibits anti-atherosclerotic effects, which are crucial in managing cardiovascular health. Atherosclerosis involves the buildup of plaques within the arteries, reducing blood flow and increasing the risk of cardiovascular events. H_2_S-induced vasodilation helps maintain adequate blood flow despite the presence of atherosclerotic plaques. Additionally, H_2_S reduces oxidative stress and inflammation, which are key contributors to plaque formation and progression. By promoting vasodilation and protecting against atherosclerosis, H_2_S can help prevent the progression of coronary artery disease and peripheral artery disease. This can reduce the risk of acute cardiovascular events such as heart attacks and strokes, offering a novel therapeutic approach to managing atherosclerotic disease [18,93,94].

Improving endothelial function is another important therapeutic mechanism of H_2_S. Endothelial dysfunction is a hallmark of many cardiovascular diseases and is characterized by a reduced ability of the endothelium to induce vasodilation. H_2_S improves endothelial function by increasing nitric oxide (NO) bioavailability and reducing oxidative stress, which is critical for proper vasodilation. Enhancing endothelial function through H_2_S therapy can help restore normal vascular responses, reduce blood pressure, and improve cardiovascular health. This is particularly beneficial in conditions such as diabetes, where endothelial dysfunction is prevalent and contributes to the development of cardiovascular complications [60,95].

H_2_S also shows promise in the treatment of pulmonary hypertension. Pulmonary hypertension is characterized by increased blood pressure in the pulmonary arteries, leading to right heart failure if left untreated. H_2_S-induced vasodilation in the pulmonary vasculature reduces pulmonary arterial pressure and alleviates the strain on the right heart. H_2_S holds potential as a treatment for pulmonary hypertension by directly lowering pulmonary vascular resistance and improving right heart function. This could improve patient outcomes and quality of life in those suffering from this condition [26].

H_2_S also has neuroprotective effects in the context of stroke. In ischemic stroke, where blood flow to the brain is compromised, H_2_S-induced vasodilation can enhance cerebral blood flow, reducing the extent of ischemic damage. Moreover, H_2_S exerts antioxidant and anti-inflammatory effects, further protecting neural tissue. The ability of H_2_S to promote vasodilation in cerebral vessels and protect against neuronal damage makes it a promising candidate for stroke therapy. Administering H_2_S during the acute phase of stroke could reduce infarct size and improve neurological outcomes [96,97,98].

Sulfurous mineral waters and peloids, rich in hydrogen sulfide (H_2_S), are known for their vasodilatory therapeutic effects. This promotes improved blood circulation and alleviates conditions related to poor vascular function. The use of H_2_S in balneotherapy has been shown to reduce blood pressure, relieve pain, and improve overall vascular health, making it a valuable tool in rehabilitative therapies aimed at enhancing cardiovascular function and managing vascular disorders [9].

## 5. Preclinical and Clinical Evidence

The therapeutic potential of H_2_S as a vasodilator is supported by a growing body of evidence from in vitro studies, preclinical animal models, and clinical trials. In vitro studies provide detailed insights into the molecular and cellular mechanisms by which H_2_S induces vasodilation. At the same time, preclinical models demonstrate their efficacy in reducing blood pressure, protecting against ischemia-reperfusion injury, preventing atherosclerosis, and managing pulmonary hypertension [99]. Clinical trials and observations are beginning to validate these findings, highlighting the potential of H_2_S as a therapeutic agent for various cardiovascular and systemic diseases.

### 5.1. In Vitro Evidence

In vitro studies have provided substantial insights into how H_2_S induces vasodilation, particularly its effects on VSMCs. Mechanistic studies using isolated VSMCs have demonstrated that H_2_S induces vasodilation primarily by activating ATP-sensitive potassium (K_ATP) channels. In vitro studies have also elucidated the cellular and molecular mechanisms by which H_2_S from therapeutic mud and sulfurous mineral waters exert its beneficial effects. These studies reveal that H_2_S promotes endothelial cell function by enhancing proliferation, migration, and tube formation, which is essential for angiogenesis. Additionally, H_2_S upregulates vascular endothelial growth factor (VEGF) and activates the PI3K/Akt pathway, improving vascular health and protecting endothelial cells from oxidative stress by activating the Nrf2 pathway. H_2_S also demonstrates significant antioxidant effects, reducing reactive oxygen species (ROS) levels and increasing the expression of antioxidant enzymes such as heme oxygenase-1 (HO-1) and superoxide dismutase (SOD) [19,32,69,100,101,102].

For fibroblasts, H_2_S-containing mud extracts enhance collagen synthesis and extracellular matrix (ECM) remodeling, which is crucial for wound healing and tissue repair. H_2_S stimulates fibroblast proliferation and migration while modulating matrix metalloproteinase (MMP) activity, ensuring balanced ECM remodeling. In inflammatory environments, H_2_S downregulates pro-inflammatory cytokines like IL-6 and TNF-α and upregulates anti-inflammatory cytokines such as IL-10 by inhibiting the NF-κB signaling pathway, thus creating a conducive environment for tissue repair [10,103].

H_2_S promotes wound healing in keratinocytes by enhancing cell proliferation and migration, leading to faster wound closure. This process involves upregulating growth factors and signaling pathways facilitating cell movement and proliferation. H_2_S also provides antioxidant protection to keratinocytes exposed to oxidative stress, such as UV-induced damage, by reducing oxidative stress markers and activating the Nrf2 pathway, which enhances the expression of antioxidant enzymes and improves cell viability [53,104].

In endothelial cells, H_2_S has been shown to enhance the activity of endothelial nitric oxide synthase (eNOS), leading to increased production of NO. This effect has been validated by in vitro assays demonstrating elevated cyclic guanosine monophosphate (cGMP) levels and subsequent vasodilation in response to H_2_S treatment [105,106,107]. Additionally, H_2_S interacts with prostacyclin signaling in endothelial cells. Specifically, in vitro studies have shown that H_2_S increases prostacyclin production by upregulating the expression of cyclooxygenase-2 (COX-2), further contributing to its overall vasodilatory effects [19,70,108].

Studies on isolated blood vessel segments, such as the aorta and mesenteric arteries, have provided further evidence of H_2_S-induced vasodilation. These isolated vessel studies demonstrate that H_2_S can significantly reduce vessel contraction in response to vasoconstrictors like phenylephrine. This highlights the role of H_2_S in modulating vascular reactivity and underscores its potential therapeutic applications in conditions where vascular tone is dysregulated [49,109,110,111].

### 5.2. Preclinical Evidence

In preclinical studies, the therapeutic potential of H_2_S has been extensively evaluated in various disease models, providing compelling evidence for its vasodilatory effects and broader cardiovascular benefits. In hypertensive rat models, administering H_2_S donors such as sodium hydrosulfide (NaHS) and GYY4137 has significantly reduced blood pressure. The antihypertensive effects of H_2_S are primarily mediated by activating ATP-sensitive potassium (K_ATP) channels, inhibiting L-type calcium channels, and reducing oxidative stress. These mechanisms, observed in both in vitro and in vivo studies, underscore the potential of H_2_S as a therapeutic agent for managing hypertension [112,113,114,115].

In models of ischemia-reperfusion injury, particularly in the context of myocardial and cerebral ischemia, H_2_S administration has demonstrated significant protective effects. These include reduced infarct size and improved tissue recovery following ischemic events. The protective effects of H_2_S are attributed to its ability to enhance blood flow during reperfusion and its antioxidant and anti-inflammatory properties. These findings from preclinical studies suggest that H_2_S could be a valuable therapeutic tool in conditions such as heart attack and stroke, where tissue recovery post-ischemia is critical [116,117,118,119].

In the context of atherosclerosis [120], preclinical studies using atherosclerosis-prone mouse models [121] have shown that H_2_S supplementation can reduce plaque formation and improve overall vascular function. These protective effects align with in vitro findings, where H_2_S has been shown to inhibit VSMC proliferation and reduce oxidative stress, which are key factors in the progression of atherosclerosis. This evidence suggests that H_2_S could play a role in preventing or slowing the progression of atherosclerotic cardiovascular disease [122,123].

Similarly, in preclinical models of pulmonary hypertension, H_2_S donors have been found to lower pulmonary arterial pressure and improve right ventricular function. These effects are consistent with the vasodilatory and protective actions of H_2_S observed in in vitro studies of pulmonary artery smooth muscle cells. The ability of H_2_S to alleviate the symptoms of pulmonary hypertension highlights its potential as a therapeutic agent in managing this challenging condition, where reducing pulmonary vascular resistance is critical to improving patient outcomes [124,125,126,127].

Animal model studies underscore the therapeutic potential of H_2_S in treating various conditions, demonstrating its anti-inflammatory, cardiovascular, wound healing, neuroprotective, metabolic, and gastrointestinal benefits. For instance, H_2_S reduces inflammation in models of carrageenan-induced paw edema and DSS-induced colitis by inhibiting NF-κB activation and decreasing leukocyte infiltration. In cardiovascular models, H_2_S donors reduce blood pressure, improve endothelial function, and protect against myocardial infarction by activating protective signaling pathways. H_2_S also accelerates wound healing in diabetic and burn injury models by promoting angiogenesis and enhancing collagen deposition [128,129].

### 5.3. Clinical Evidence

Clinical research has increasingly validated the therapeutic potential of H_2_S in managing various cardiovascular and systemic conditions. In the context of hypertension, clinical trials involving H_2_S donors such as sodium thiosulfate have shown promising results in lowering blood pressure in hypertensive patients. These clinical findings align with in vitro and preclinical evidence, demonstrating the vasodilatory effects of H_2_S on vascular smooth muscle cells. The ability of H_2_S to reduce vascular resistance and enhance blood flow underpins its potential as an effective treatment for hypertension [130].

Cardioprotection in myocardial ischemia has also been a key focus of clinical investigations. In patients undergoing percutaneous coronary interventions (PCI) for myocardial infarction, adjunctive therapy with H_2_S donors has been observed to reduce myocardial damage [131]. These clinical benefits are consistent with preclinical and in vitro data, which highlight the role of H_2_S in improving blood flow, reducing oxidative stress, and limiting tissue damage during ischemic events. Such findings suggest that H_2_S could be a valuable adjunct in treating acute myocardial infarction, protecting against reperfusion injury [132,133,134,135,136].

Human studies have shown that elevated levels of endogenous H_2_S are associated with improved vascular function in atherosclerosis and endothelial dysfunction. Moreover, the administration of exogenous H_2_S has enhanced endothelial function in patients with coronary artery disease, reflecting the mechanisms observed in earlier in vitro and preclinical studies. These observations underscore the potential of H_2_S as a therapeutic agent in managing atherosclerotic disease and preventing its progression [137,138].

For pulmonary hypertension, early clinical studies, including case reports and small trials, suggest that H_2_S donors may reduce pulmonary arterial pressure and alleviate symptoms in patients. These outcomes are consistent with the vasodilatory effects of H_2_S observed in preclinical models of pulmonary hypertension, where H_2_S has been shown to lower pulmonary vascular resistance and improve right ventricular function. This evidence supports the further exploration of H_2_S-based therapies in treating pulmonary hypertension [139,140,141].

Neuroprotection in stroke is another area where H_2_S is being actively investigated [142]. Ongoing clinical trials are evaluating the efficacy of H_2_S in acute ischemic stroke, with preliminary data suggesting potential benefits in improving cerebral blood flow and reducing brain injury [143]. These effects mirror those observed in preclinical models and studies of isolated cerebral vessels, where H_2_S has been shown to enhance blood flow and protect against ischemic damage [19]. The results of these trials could pave the way for using H_2_S as a neuroprotective agent in stroke therapy [144].

Beyond these specific conditions, broader clinical research highlights the versatility of H_2_S in treating various health issues [145]. In cardiovascular diseases, H_2_S donors [146] have not only reduced blood pressure and improved vascular function in hypertensive patients but have also enhanced microcirculation and reduced claudication pain in individuals with peripheral artery disease [147]. In wound healing, H_2_S-releasing dressings have accelerated the healing of diabetic ulcers and burn wounds, promoting faster epithelialization and reducing scar formation [148,149,150]. Inflammatory conditions, such as rheumatoid arthritis [151,152,153] and inflammatory bowel disease, have also benefited from H_2_S therapy, which has been shown to reduce inflammatory markers and improve clinical outcomes [154].

Furthermore, H_2_S demonstrates significant potential in metabolic disorders, including type 2 diabetes and non-alcoholic fatty liver disease [155]. Clinical studies indicate that H_2_S can improve glycemic control, enhance insulin sensitivity, and support liver function, making it a promising candidate for managing these conditions [156]. Overall, these findings highlight the therapeutic promise of H_2_S and support its integration into clinical practice for a wide range of health conditions, emphasizing its role as a versatile and effective treatment modality [157].

## 6. Discussion and Future Directions and Challenges

While significant progress has been made in understanding the therapeutic potential of H_2_S, several research gaps must be addressed to optimize H_2_S-based therapies and ensure their effective translation from preclinical findings to clinical treatments [158].

First, it is essential to elucidate the molecular mechanisms through which H_2_S exerts its effects. This includes understanding the specific signaling pathways, such as those involving ion channels, enzymes, and transcription factors, and how H_2_S modulates these pathways under different physiological and pathological conditions. Further studies on post-translational modifications, particularly S-sulfhydration of proteins, are crucial to uncover the functional implications of these modifications in H_2_S signaling [19,26,32].

Second, comprehensive pharmacokinetic and pharmacodynamic studies are necessary to understand better the bioavailability, metabolism, and long-term effects of H_2_S donors. This includes determining the optimal routes of administration and dosing regimens to achieve sustained therapeutic effects without causing adverse effects. Long-term safety studies are critical to assess potential desensitization, toxicity, and changes in endogenous H_2_S production with chronic administration [148].

Third, safety and toxicity assessments are critical for advancing H_2_S-based therapies. These studies should establish the therapeutic window and safety margins for various H_2_S donors, identifying doses that provide therapeutic benefits without causing harm. Evaluating organ-specific toxicities, especially in the liver, kidneys, and cardiovascular system, is vital to ensure comprehensive safety profiles. Understanding the dose-response relationship and potential organ-specific effects will help develop safer and more effective therapies [159].

Fourth, research into disease-specific applications of H_2_S is essential to tailor therapies for maximum efficacy in treating conditions such as cardiovascular diseases, neurodegenerative diseases, metabolic disorders, and cancer. For example, investigating how H_2_S influences cardiac remodeling, contractility, and electrical activity can lead to targeted heart failure and arrhythmias therapies. Similarly, detailed studies on the neuroprotective mechanisms of H_2_S can advance treatments for Alzheimer’s disease, Parkinson’s disease, and stroke [80,160].

Lastly, technological advancements are pivotal in enhancing the study and application of H_2_S in therapeutics. Innovations in delivery systems, such as nanoparticles, hydrogels, and implantable devices, can improve the targeted and sustained release of H_2_S [161]. Advanced analytical techniques, including high-resolution mass spectrometry and fluorescent probes, can provide deeper insights into H_2_S biology [62,162,163]. Gene editing technologies like CRISPR/Cas9 and personalized medicine approaches can also optimize endogenous H_2_S production and treatment efficacy [164]. Addressing these research gaps through collaborative efforts will unlock the full therapeutic potential of H_2_S, leading to effective and safe treatments for a wide range of medical conditions. The future outlook for integrating H_2_S signaling insights into clinical practice is promising. Advances in research, technological innovations, and clinical trials pave the way for developing and implementing H_2_S-based therapies. The therapeutic potential of mud and sulfurous mineral waters, enriched with H_2_S, offers a unique and natural approach to enhancing health and treating various conditions. By continuing to explore and harness the benefits of H_2_S, the medical community can develop novel and effective treatments that improve patient outcomes and advance the field of medicine [32,156,165].

## 7. Conclusions

The comprehensive understanding of hydrogen sulfide (H_2_S) signaling highlights its vast therapeutic potential, especially in inflammation, oxidative stress, vascular health, and tissue repair. Current and future clinical trials aim to expand the indications for H_2_S-based therapies, optimizing dosing regimens and delivery methods to ensure safety and efficacy. Streamlined regulatory pathways and developing clinical practice guidelines are essential to integrate H_2_S therapies into standard medical care, ultimately improving patient outcomes across various conditions such as cardiovascular diseases, diabetes, neurodegenerative disorders, and inflammatory diseases.

Advancements in H_2_S research will significantly benefit personalized medicine. Identifying biomarkers for H_2_S activity and production will enable tailored therapies, ensuring that treatments match individual patient profiles. Genetic profiling can enhance this approach by pinpointing patients most likely to benefit from H_2_S-based interventions. Technological innovations in drug delivery, such as nanoparticle-based systems, hydrogels, and non-invasive methods like transdermal patches and inhalation therapies, promise to enhance the targeted delivery and sustained release of H_2_S, improving therapeutic outcomes and patient compliance.

The therapeutic use of mud and sulfurous mineral waters, rich in H_2_S, represents a natural extension of these findings. Their incorporation into hospitals and rehabilitation centers can support the recovery and management of chronic diseases, offering holistic benefits and improving overall patient care. The therapeutic potential of mud and sulfurous mineral waters is closely linked to their H_2_S content. Understanding the mechanisms of H_2_S release and action in these natural therapies can enhance their clinical applications.

## Figures and Tables

**Figure 1 antioxidants-13-01158-f001:**
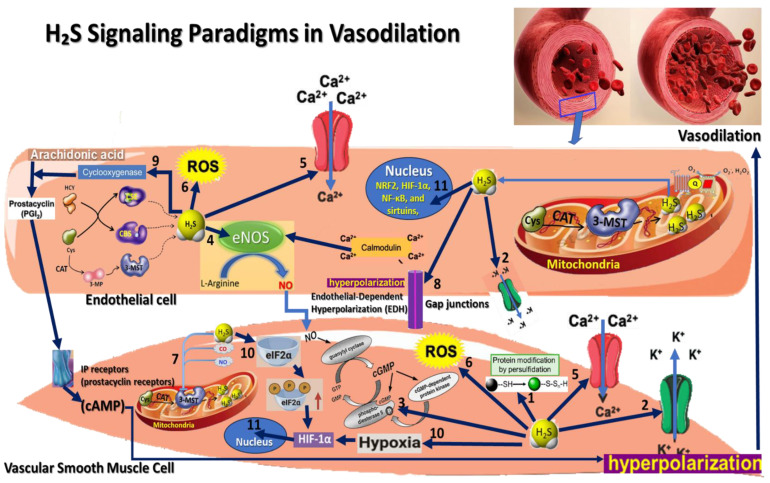
This figure illustrates the various molecular and signaling paradigms through which H_2_S contributes to vasodilation: 1. Persulfidation Signaling—a post-translational modification where a sulfur atom is added to the thiol (-SH) group of cysteine residues, forming persulfide (-SSH) groups; 2. K_ATP Channel Activation Paradigm—reduces calcium influx, resulting in smooth muscle relaxation and vasodilation; 3. cGMP Pathway Activation Paradigm—elevated cGMP activates protein kinase G (PKG), which decreases intracellular calcium levels, causing smooth muscle relaxation and vasodilation; 4. Endothelial Nitric Oxide Synthase (eNOS) activation Paradigm; 5. Calcium Signaling Modulation Paradigm; 6. Redox Signaling and Antioxidant Effects Paradigm; 7. Interaction with Other Gasotransmitters Paradigm; 8. Endothelial-Dependent Hyperpolarization (EDH) Paradigm; 9. Interaction with Prostacyclin Signaling Paradigm; 10. Hypoxia Response Paradigm; 11. H_2_S interaction with various transcription factors.

**Table 1 antioxidants-13-01158-t001:** The specific keyword combinations used for searching relevant scientific articles.

Keywords	PubMed	Scopus	Web of Science	Total
“Hydrogen sulfide” [Title] AND “vasodilation” [Title]	10	2	7	19
“H_2_S” [Title] AND “vasodilation” [Title]	4	0	4	8
“Hydrogen sulfide” [Title] AND “vascular regulation” [Title]	2	6	7	15
“H_2_S” [Title] AND “vascular regulation” [Title]	0	0	7	7

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
