# Peer review of "Signaling Paradigms of H2S-Induced Vasodilation: A Comprehensive Review"

_antioxidants, 2024, doi:10.3390/antiox13101158_

Round 1

Reviewer 1 Report

This manuscript encompasses a systematic literature review that explores the intricate mechanisms underlying Hâ‚‚S-induced vasodilation. The results suggest that signaling paradigms of Hâ‚‚S-induced vasodilation offer valuable insights into its physiological role and therapeutic potential, promising innovative approaches for managing various vascular diseases through the modulation of vascular tone.  The results are of interest to readers. However, weaknesses of the paper need to be improved by the authors for possible publication in journal Antioxidants.

1.  P. 2, lines 90-93 “Endogenously produced in the human body through enzymatic pathways involving 90 cystathionine β-synthase (CBS), cystathionine γ-lyase (CSE), and 3-mercaptopyruvate sul-91 furtransferase (3-MST) [25], Hâ‚‚S is present in various tissues, including the brain, liver, kidneys, and blood vessels [26].” should be revised as two sentences “Endogenously produced in the human body through enzymatic pathways involving cystathionine β-synthase (CBS), cystathionine γ-lyase (CSE), and 3-mercaptopyruvate sul-91 furtransferase (3-MST) [25]. Hâ‚‚S is present in various tissues, including the brain, liver, 92 kidneys, and blood vessels [26].”  

2. Abbreviations should be defined only in the first present in the text. For example, in p5, lines 222-228 “Hâ‚‚S inhibits phosphodiesterase (PDE) enzymes, increasing smooth muscle cells' cyclic GMP (cGMP) levels. Elevated cGMP activates protein kinase G (PKG), which reduces intracellular calcium levels and causes smooth muscle relaxation and vasodilation. The cGMP pathway activation paradigm is another key mechanism through which Hâ‚‚S induces vasodilation. Hâ‚‚S significantly modulates cyclic guanosine monophosphate (cGMP) levels within VSMCs. This process begins with Hâ‚‚S inhibiting the activity of phosphodiesterase (PDE) enzymes, …..” should be corrected as “Hâ‚‚S inhibits phosphodiesterase (PDE) enzymes, increasing smooth muscle cells' cyclic GMP (cGMP) levels. Elevated cGMP activates protein kinase G (PKG), which reduces intracellular calcium levels and causes smooth muscle relaxation and vasodilation. The cGMP pathway activation paradigm is another key mechanism through which Hâ‚‚S induces vasodilation. Hâ‚‚S significantly modulates cGMP levels within VSMCs. This process begins with Hâ‚‚S inhibiting the activity of PDE enzymes, …..”. There are many such kind errors in the manuscript. Please check and correct the incorrected abbreviations thoroughly in the text.

3. Figure 1 legend is not enough to understand the figures. Detailed description is required for understanding the figure contents and results.

 4. P. 8, lines 352-354, “In addition to these direct effects, Hâ‚‚S may also influence the production or action of other endothelium-derived hyperpolarizing factors (EDHFs), such as epoxyeicosatrienoic acids (EETs) or hydrogen peroxide (Hâ‚‚Oâ‚‚). These factors further contribute to the hyperpolarization of VSMCs and enhance vasodilation.”  The description is not clear since hydrogen peroxide (Hâ‚‚Oâ‚‚) is a free radical which causes opposing effect of vasodilation.  More detailed description is required for understanding as “Enhanced production of endothelium-derived hyperpolarizing factors and reduced production of hydrogen peroxide (Hâ‚‚Oâ‚‚) induced by Hâ‚‚S contribute to the hyperpolarization of VSMCs and enhance vasodilation, respectively.”

5. P. 11, line 520, Preclinical and Clinical Evidence should be 5. Preclinical and Clinical Evidence. Subsequently, “5. Discussion and Future Directions and Challenges, and 6. Conclusions” should be modified as “6. Discussion and Future Directions and Challenges, and 7. Conclusions”.  

Author Response

We sincerely appreciate the reviewer's constructive feedback on our manuscript. The comments have been incredibly helpful in enhancing the clarity and rigor of our work. Below, we address your concerns point by point and outline the revisions made to the manuscript.

  1. ”P. 2, lines 90-93 “Endogenously produced in the human body through enzymatic pathways involving 90 cystathionine β-synthase (CBS), cystathionine γ-lyase (CSE), and 3-mercaptopyruvate sul-91 furtransferase (3-MST) [25], Hâ‚‚S is present in various tissues, including the brain, liver, kidneys, and blood vessels [26].” should be revised as two sentences “Endogenously produced in the human body through enzymatic pathways involving cystathionine β-synthase (CBS), cystathionine γ-lyase (CSE), and 3-mercaptopyruvate sul-91 furtransferase (3-MST) [25]. Hâ‚‚S is present in various tissues, including the brain, liver, 92 kidneys, and blood vessels [26].” - The original sentence is somewhat lengthy and combines two distinct pieces of information—(1) the enzymatic pathways involved in endogenous Hâ‚‚S production, and (2) the tissues where Hâ‚‚S is present. The reviewer recommends separating this into two sentences to improve clarity, creating a more balanced presentation of information. Here's the revised version: “Endogenously produced in the human body through enzymatic pathways involving cystathionine β-synthase (CBS), cystathionine γ-lyase (CSE), and 3-mercaptopyruvate sulfurtransferase (3-MST) [25]. Hâ‚‚S is present in various tissues, including the brain, liver, kidneys, and blood vessels [26].”
  2. ”Abbreviations should be defined only in the first present in the text. For example, in p5, lines 222-228 “Hâ‚‚S inhibits phosphodiesterase (PDE) enzymes, increasing smooth muscle cells' cyclic GMP (cGMP) levels. Elevated cGMP activates protein kinase G (PKG), which reduces intracellular calcium levels and causes smooth muscle relaxation and vasodilation. The cGMP pathway activation paradigm is another key mechanism through which Hâ‚‚S induces vasodilation. Hâ‚‚S significantly modulates cyclic guanosine monophosphate (cGMP) levels within VSMCs. This process begins with Hâ‚‚S inhibiting the activity of phosphodiesterase (PDE) enzymes, …..” should be corrected as “Hâ‚‚S inhibits phosphodiesterase (PDE) enzymes, increasing smooth muscle cells' cyclic GMP (cGMP) levels. Elevated cGMP activates protein kinase G (PKG), which reduces intracellular calcium levels and causes smooth muscle relaxation and vasodilation. The cGMP pathway activation paradigm is another key mechanism through which Hâ‚‚S induces vasodilation. Hâ‚‚S significantly modulates cGMP levels within VSMCs. This process begins with Hâ‚‚S inhibiting the activity of PDE enzymes, …..”. There are many such kind errors in the manuscript. Please check and correct the incorrected abbreviations thoroughly in the text.” - The reviewer has suggested correcting this redundancy, and similar errors should be addressed throughout the document. We checked the manuscript for this issue.
  3. ”Figure 1 legend is not enough to understand the figures. Detailed description is required for understanding the figure contents and results.” - We agree that the legend should provide a more comprehensive description. The legend for Figure 1 has been revised to include a detailed explanation of the figure contents to ensure better understanding.
  4. P. 8, lines 352-354, “In addition to these direct effects, Hâ‚‚S may also influence the production or action of other endothelium-derived hyperpolarizing factors (EDHFs), such as epoxyeicosatrienoic acids (EETs) or hydrogen peroxide (Hâ‚‚Oâ‚‚). These factors further contribute to the hyperpolarization of VSMCs and enhance vasodilation.”  The description is not clear since hydrogen peroxide (Hâ‚‚Oâ‚‚) is a free radical which causes opposing effect of vasodilation.  More detailed description is required for understanding as “Enhanced production of endothelium-derived hyperpolarizing factors and reduced production of hydrogen peroxide (Hâ‚‚Oâ‚‚) induced by Hâ‚‚S contribute to the hyperpolarization of VSMCs and enhance vasodilation, respectively.” - Thank you for this important observation. We have revised the sentence to clarify that Hâ‚‚S reduces Hâ‚‚Oâ‚‚ production, contributing to vasodilation. The revised text: “Enhanced production of endothelium-derived hyperpolarizing factors and reduced production of hydrogen peroxide (Hâ‚‚Oâ‚‚) induced by Hâ‚‚S contribute to the hyperpolarization of VSMCs and enhance vasodilation.”
  5. P. 11, line 520, Preclinical and Clinical Evidence should be 5. Preclinical and Clinical Evidence. Subsequently, “5. Discussion and Future Directions and Challenges, and 6. Conclusions” should be modified as “6. Discussion and Future Directions and Challenges, and 7. Conclusions”. - We appreciate this formatting suggestion and have revised the section titles as recommended.

We are grateful for the constructive review, and we hope the revised manuscript meets the reviewer's expectations.

Reviewer 2 Report

My concern is that the authors just described that H2S S-sulfurates cysteine residues. It is not correct. H2S S-sulfurates oxidized cysteine residues, while polysulfides including H2Sn S-sulfurate cysteine residues (Kabil et al., 2014).

1. 3.1. line 181

     H2Sn and other per- and poly-sulfides S-sulfurate thiols but not H2S.

It is to be corrected.

2. 3.7. line 312

The synergistic effect of H2S with NO in vasculature was initially reported by Hosoki et al (1997), and later Whiteman et al (2006) suggested that the chemical interaction of both molecules produces nitrosothiol. It was later proposed as HNO (eberhardt et al., 2014), SSNO- (Cortese-Krott et al., 2015), H2Sn (Miyamoto et al., 2017). It is to be added.

3. Some of references miss Journal name. For example, 9, 37, 52 and so on. It should be added.

Author Response

We sincerely thank the reviewer for the insightful comments on our manuscript. The suggestions are precious and have contributed significantly to improving the accuracy and clarity of our work. Below, we provide a detailed response to each of the comments.

Major Comment: “My concern is that the authors just described that H2S S-sulfurates cysteine residues. It is not correct. H2S S-sulfurates oxidized cysteine residues, while polysulfides including H2Sn S-sulfurate cysteine residues (Kabil et al., 2014).”

We agree with the reviewer’s correction and have amended the manuscript accordingly. The statement in Section 3.1 that previously mentioned Hâ‚‚S S-sulfurating cysteine residues has been revised to clarify that Hâ‚‚S S-sulfurates oxidized cysteine residues, and polysulfides, including Hâ‚‚Sn, are responsible for S-sulfurating reduced cysteine residues. This correction aligns our manuscript with the established literature, including Kabil et al. (2014).

Detailed Comments:

  1. Comment on Line 181: "Hâ‚‚Sn and other per- and poly-sulfides S-sulfurate thiols but not Hâ‚‚S."

We have corrected the statement in the manuscript to accurately reflect the role of polysulfides, including Hâ‚‚Sn, in S-sulfurating thiols, while Hâ‚‚S does not perform this function. 

  1. Comment on Line 312: The reviewer requested additional references on the synergistic effect of Hâ‚‚S with NO, citing necessary studies by Hosoki et al. (1997), Whiteman et al. (2006), Eberhardt et al. (2014), Cortese-Krott et al. (2015), and Miyamoto et al. (2017).

We appreciate this valuable suggestion and have incorporated the requested references to provide a more comprehensive discussion on the Hâ‚‚S-NO interaction. The section has been expanded to include the contributions of these studies, outlining the progression of research on the synergistic effects of Hâ‚‚S and NO in the vasculature.

  1. Missing Journal Names in References: The reviewer noted that some references were missing journal names, including references 9, 37, and 52.

We have thoroughly reviewed the reference section and added the missing journal names to references 9, 37, 52, and all other incomplete references.